

# SCMS
## Shooting Club Management System



**Autorzy**: Daniel Börner · Julia Kroczek · Mateusz Polito · Krzysztof Otręba

**Opiekun:** Marcin Pietranik

**Streszczenie**

Projekt Shooting Club Management System odpowiada na rosnące zapotrzebowanie na cyfryzację procesów zarządzania w klubach strzeleckich w Polsce. System oferuje kompleksowe rozwiązanie do zarządzania klubem strzeleckim, organizacji zawodów oraz automatyzacji procesów administracyjnych. Głównym celem projektu jest zwiększenie efektywności operacyjnej klubów strzeleckich poprzez digitalizację dokumentacji, automatyzację zapisów na treningi i zawody oraz usprawnienie procesu obliczania wyników zawodów. Wdrożenie systemu pozwala na znaczącą redukcję kosztów operacyjnych, minimalizację błędów ludzkich oraz poprawę jakości obsługi członków klubu. Projekt został zrealizowany z wykorzystaniem nowoczesnych technologii webowych, zapewniając intuicyjny interfejs użytkownika oraz niezawodne działanie systemu. Zastosowane rozwiązania technologiczne umożliwiają tanie wdrożenie systemu w wielu klubach naraz oraz zapewniają skalowalność systemu wraz ze wzrostem liczby użytkowników.

## 1 WSTĘP

### 1.1 Opis problemu

Obecnie w Polsce większość klubów strzeleckich nie korzysta z dedykowanych rozwiązań informatycznych, co prowadzi do opóźnień oraz nieefektywności w zarządzaniu działalnością klubu i organizacji zawodów. Dokumentacja klubowa często prowadzona jest w formie papierowej, a wynik zawodnika w danej konkurencji zawodów obliczany jest ręcznie na tzw. metryczkach (przykład na Rysunku 1), co zwiększa ryzyko błędów i utrudnia sprawną obsługę uczestników oraz przejrzystość procesu. Brak zautomatyzowanego systemu zarządzania negatywnie wpływa na funkcjonowanie klubów, ich wizerunek oraz satysfakcję członków i uczestników zawodów.

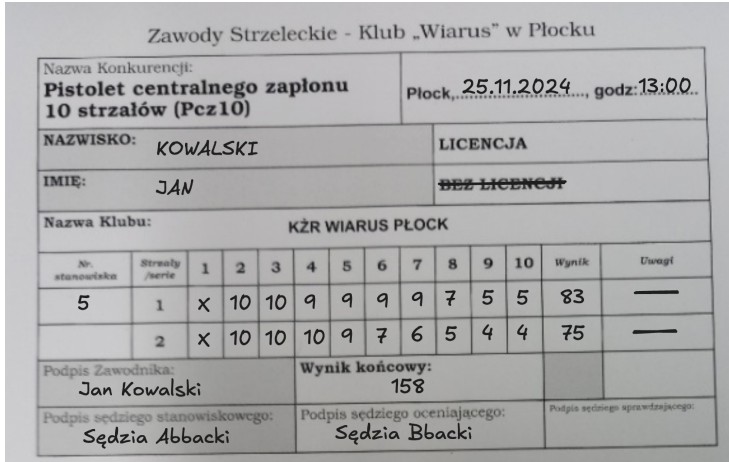

Rysunek 1: Przykład wypełnionej metryczki [Źródło: opracowanie własne].

### 1.2 Cele projektu

Głównym zamierzeniem projektu jest projekt i implementacja systemu informatycznego wspierającego zarządzanie klubem strzeleckim. W kontekście technicznym ma to na celu:

- Automatyzację procesów administracyjnych związanych z członkostwem i harmonogramami.
- Ułatwienie organizacji zawodów i treningów strzeleckich, w tym obsługę zapisów.

- Zarządzanie konkurencjami, umożliwiające tworzenie własnych metryczek i indywidualnych sposobów obliczania wyników.

- Automatyczne obliczanie wyników dla zawodników w poszczególnych konkurencjach.

- Umożliwienie monitorowania postępów w zawodach strzeleckich poprzez szczegółowe statystyki.

- Automatyczne generowanie raportów z zawodów.

- Stworzenie intuicyjnego interfejsu użytkownika, który pozwoli osobom o różnym poziomie zaawansowania technologicznego korzystać z systemu.

W kontekście biznesowym projekt ma na celu:

- Zwiększenie efektywności operacyjnej klubów strzeleckich poprzez oszczędność czasu i redukcję kosztów.

- Poprawa doświadczenia użytkowników, zarówno administratorów i członków klubu, jak i uczestników zawodów.

- Zwiększenie transparentności i wiarygodności wyników zawodów.

- Podniesienie zaangażowania i satysfakcji uczestników zawodów strzeleckich.

- Wzmocnienie wizerunku klubu jako nowoczesnej i profesjonalnej organizacji.

## 2  PRACE ZWIĄZANE Z TEMATEM

### 2.1  Analiza istniejących rozwiązań i technologii

Większość polskich klubów strzeleckich nie korzysta z zaawansowanych systemów do zarządzania. Praca administracyjna opiera się często na ręcznym wypełnianiu dokumentów i tabel w arkuszach kalkulacyjnych, co wymaga dużego nakładu czasu i pracy, prowadząc do potencjalnych błędów w organizacji zawodów i zarządzaniu danymi. Po analizie rynku można stwierdzić, że w tej niszy istnieją dwie warte uwagi aplikacje.

Pierwsza z nich to Strzelnica.app (`https://strzelnicaapp.pl/`) - aplikacja przeznaczona dla właścicieli klubów, skupiona na zarządzaniu zawodami strzeleckimi. Pozwala na utworzenie zawodów, zapisanie zawodników przez e-mail oraz wpisywanie wyników i śledzenie ich na bieżąco na ekranie. SCMS zdecydowanie przewyższa Strzelnica.app pod względem funkcjonalności i nowoczesności. Nasz system integruje pełne zarządzanie klubem, obejmując członków, dokumentację, treningi i zawody, podczas gdy Strzelnica.app oferuje jedynie ograniczone możliwości zarządzania zawodami. SCMS automatyzuje procesy takie jak rezerwacje osi strzeleckich, rejestracja członków i generowanie raportów, a także posiada zaawansowane funkcje, takie jak tworzenie metryczek z kodami QR, automatyczne wyliczanie wyników oraz możliwość dostosowywania formuł za pomocą generatora. Dodatkowo, SCMS oferuje nowoczesny design i wszystkie funkcje w aplikacji przeglądarkowej, podczas gdy Strzelnica.app ogranicza funkcjonalność swojej wersji przeglądarkowej, a aplikacja desktopowa cechuje się przestarzałym wyglądem. Co więcej, SCMS umożliwia zakładanie kont użytkowników z historią wyników i statystykami, podczas gdy w Strzelnica.app zawodnicy muszą każdorazowo rejestrować się przez prosty formularz, wprowadzając wszystkie dane od nowa, bez możliwości śledzenia swoich postępów czy historii zawodów.

Drugim rozwiązaniem jakie można znaleźć jest CelPal (`https://celpal.net/`) - system przeznaczony dla administracji klubów strzeleckich. CelPal to aplikacja skupiająca się wyłącznie na podstawowej administracji klubu, takiej jak elektroniczna książka rejestru pobytu, rejestracja klubowiczów czy karty członkowskie RFID, jednak nie oferuje żadnych funkcji związanych z organizacją zawodów czy zarządzaniem treningami. Produkt ten wydaje się obecnie niedostępny i nierozwijany, co dodatkowo ogranicza jego przydatność. W porównaniu do CelPal, SCMS zapewnia kompleksowe wsparcie dla klubów, integrując zarządzanie treningami, zawodami, członkami i raportami w jednym nowoczesnym systemie dostępnym w pełni przez aplikację przeglądarkową.

### 2.2  Wybór technologii

Django (`https://www.djangoproject.com/`) zostało wybrane jako framework backendowy dla tego projektu ze względu na swoje liczne zalety, które idealnie odpowiadają potrzebom dynamicznych aplikacji webowych. Django, napisane w języku Python, umożliwia szybkie prototypowanie i wysoką produktywność dzięki bogatemu zestawowi wbudowanych narzędzi, takich jak mechanizmy walidacji danych, obsługa sesji czy system migracji bazy danych. Jego modularna struktura pozwala na łatwe rozszerzanie

funkcjonalności, a wbudowany system migracji umożliwia szybkie wprowadzanie zmian w strukturze danych, co jest szczególnie istotne w projektach rozwijanych zgodnie ze zwinnymi metodologiami. Framework ten wspiera tworzenie aplikacji w oparciu o architekturę MVC (Model-View-Controller), co zapewnia czytelność kodu i ułatwia rozwój projektu. Django wyróżnia się także szeroką gamą bibliotek, które pozwalają efektywnie wdrożyć takie funkcjonalności jak zaawansowane uwierzytelnianie czy integracje z zewnętrznymi serwisami. Dzięki temu framework oferuje elastyczność, skalowalność oraz możliwość łatwej integracji z innymi technologiami, co czyni go idealnym wyborem dla backendu aplikacji.

Frontend został zaprojektowany z wykorzystaniem React (`https://react.dev/`), Tailwind CSS (`https://tailwindcss.com/`) oraz biblioteki shadcn/ui (`https://ui.shadcn.com/`). React wspiera dynamiczne, komponentowe tworzenie interfejsu użytkownika, co ułatwia rozwój i utrzymanie aplikacji. Tailwind CSS zapewnia elastyczne tworzenie nowoczesnych i responsywnych elementów wizualnych. Z kolei biblioteka shadcn/ui dostarcza gotowe, stylowe komponenty oparte na Tailwind CSS, co przyspiesza tworzenie spójnych i estetycznych interfejsów użytkownika.

PostgreSQL (`https://www.postgresql.org/`) został wybrany jako system zarządzania bazą danych ze względu na swoje kluczowe cechy, które idealnie odpowiadają potrzebom aplikacji. Obsługa formatu JSON pozwala na przechowywanie nieregularnych danych, takich jak szczegóły konkurencji na zawodach, a relacyjny charakter bazy zapewnia integralność danych dzięki kluczom głównym, obcym i więzom integralności. PostgreSQL oferuje wysoką wydajność i skalowalność, co czyni go niezawodnym rozwiązaniem dla aplikacji obsługujących dużą liczbę użytkowników. Dodatkowo zaawansowane możliwości zapytań umożliwiają efektywne przetwarzanie nawet skomplikowanych operacji na danych.

W projekcie wykorzystano również inne technologie wspierające działanie systemu, takie jak Docker (`https://www.docker.com/`) i Redis (`https://redis.io/`). Docker umożliwia łatwe wdrażanie systemu w różnych środowiskach oraz wspiera pracę zespołową, zapewniając spójność między środowiskami programistycznymi. Redis został zastosowany jako narzędzie do buforowania danych razem z Celery (`https://docs.celeryq.dev/`) jako kolejką zadań.

Wszystkie wybrane technologie – React.js, Django oraz PostgreSQL – są dostępne na licencjach open source, co oznacza, że można z nich korzystać bez żadnych opłat licencyjnych. Dzięki temu projekt może być realizowany bez ponoszenia dodatkowych kosztów związanych z oprogramowaniem. Dodatkowo, open source oznacza pełen dostęp do kodu źródłowego, co pozwala na dowolne dostosowanie narzędzi do specyficznych wymagań projektu. Aktywne społeczności użytkowników i deweloperów wspierające każdą z technologii zapewniają również dostęp do obszernej dokumentacji, przykładów oraz gotowych rozwiązań, co znacząco przyspiesza rozwój aplikacji i rozwiązywanie potencjalnych problemów. Dzięki temu wybrane technologie stanowią solidną, elastyczną i ekonomiczną podstawę dla realizacji projektu.

## 2.3 Zasoby, podział prac, ograniczenia i trudności

Projekt realizowany był w warunkach ograniczeń czasowych i zasobowych, co wymagało intensywnej pracy oraz elastyczności w podejściu do podziału zadań. Rozpoczął się na początku października, a termin zakończenia wyznaczono na pierwszą połowę grudnia, co dawało około dwóch miesięcy na realizację. Krótki czas trwania projektu wymagał efektywnego zarządzania zadaniami, aby dostarczyć w pełni funkcjonalny system w wyznaczonym terminie.

Zespół projektowy składał się z czterech osób: dwóch odpowiedzialnych za backend i dwóch za frontend. Pod koniec listopada prace nad backendem zostały zakończone, jednak rozwój frontendu zaczął pozostawać w tyle. Aby przyspieszyć postęp, kiedy backend był już w fazie ukończenia, jedna osoba z zespołu backendowego została przeniesiona do pracy nad frontendem aby poprawić tempo jego realizacji.

W trakcie projektu napotkano również kilka problemów technicznych i organizacyjnych. Tworzenie systemu metryczek dla zawodników okazało się wyjątkowo złożone i wymagało ponad miesiąca pracy jednej osoby z zespołu frontendowego, co opóźniło inne zadania związane z interfejsem użytkownika. Ograniczony czas oraz zasoby zespołu wymusiły priorytetyzację funkcji systemu, co wiązało się z rezygnacją z części mniej istotnych rozwiązań. Dodatkowo, współpraca między backendem a frontendem wymagała dodatkowej synchronizacji w późniejszych etapach projektu, kiedy backend był już ukończony, a frontend nadal wymagał znacznego nakładu pracy.

Podsumowując, pomimo trudności związanych z krótkim czasem trwania projektu i ograniczonymi zasobami, zespół wykazał się efektywnym dostosowaniem do zmieniających się okoliczności. Dzięki temu możliwe było zrealizowanie kluczowych funkcji systemu w założonym terminie.

## 3 WYNIKI

W ramach realizacji projektu Shooting Club Management System opracowano kompleksową aplikację webową, która znacząco transformuje sposób zarządzania klubami strzeleckimi. System został podzie-

lony na cztery główne moduły funkcjonalne, każdy odpowiadający za kluczowe aspekty działalności klubu.

## 3.1 Moduł zarządzania klubem strzeleckim

Podstawowy moduł systemu odpowiedzialny za zarządzanie fundamentalnymi aspektami działalności klubu. Umożliwia:

- Zarządzanie podstawowymi danymi o klubie
- Prowadzenie elektronicznej ewidencji członków klubu
- Zarządzanie statusami członkowskimi i uprawnieniami
- Automatyzację procesu komunikacji z członkami klubu
- Zarządzanie osiami strzeleckimi należących do klubu

## 3.2 Moduł zarządzania licencjami

Moduł dedykowany zarządzaniu uprawnieniami i licencjami w klubie, który obejmuje:

- System zarządzania licencjami sędziowskimi, w tym:
  - Monitorowanie terminów ważności licencji
  - Śledzenie uprawnień sędziowskich
  - Przydzielanie sędziemu nowych licencji sędziowskich
- System zarządzania licencjami zawodniczymi:
  - Monitoring terminów odnowienia licencji
  - Przydzielanie zawodnikowi nowych licencji zawodniczych
  - System weryfikacji danych do dodanej licencji

## 3.3 Moduł zarządzania treningami strzeleckimi

Komponent odpowiedzialny za organizację i zarządzanie treningami strzeleckimi, oferujący:

- Zaawansowany system zarządzania harmonogramem treningów
- Automatyczny system zapisów na treningi z uwzględnieniem dostępności miejsc na daną godzinę
- Monitoring obłożenia strzelnicy
- System powiadomień oraz potwierdzeń dla uczestników treningów

## 3.4 Moduł zarządzania zawodami strzeleckimi

Najbardziej rozbudowany moduł systemu, zapewniający kompleksową obsługę zawodów strzeleckich. Oferuje on:

- Zaawansowany system tworzenia szablonów konkurencji strzeleckich (Zrzut ekranu na Rysunku 2) pozwalający na dowolne dostosowanie wyglądu wirtualnej metryczki. Metryczkę wirtualną tworzy się poprzez formowanie siatki dwuwymiarowej z różnych typów danych (Dostępne typy to: liczbowy, liczbowy z uwzględnieniem strzałów centralnych, czasowy, zero-jedynkowy, tekstowy, oraz typ pusty). Następnie z tych typów danych układa się wzór matematyczny, który będzie wykorzystywany do obliczenia wyniku końcowego. Co więcej, system udostępnia możliwość dodawania dodatkowych wyrażeń matematycznych, które służą informatywnemu wyświetlaniu uzupełniających danych w tabeli z rankingiem (Przykład na Rysunku 3). Na przykładowym zrzucie ekranu (Rysunek 2) można zobaczyć konkurencję strzelecką, której wynik to liczba trafień pomnożona przez 10, następnie wynik podzielony przez sumę czasu, dodatkowego czasu karnego oraz liczby nietrafień pomnożony przez 5 sekund.

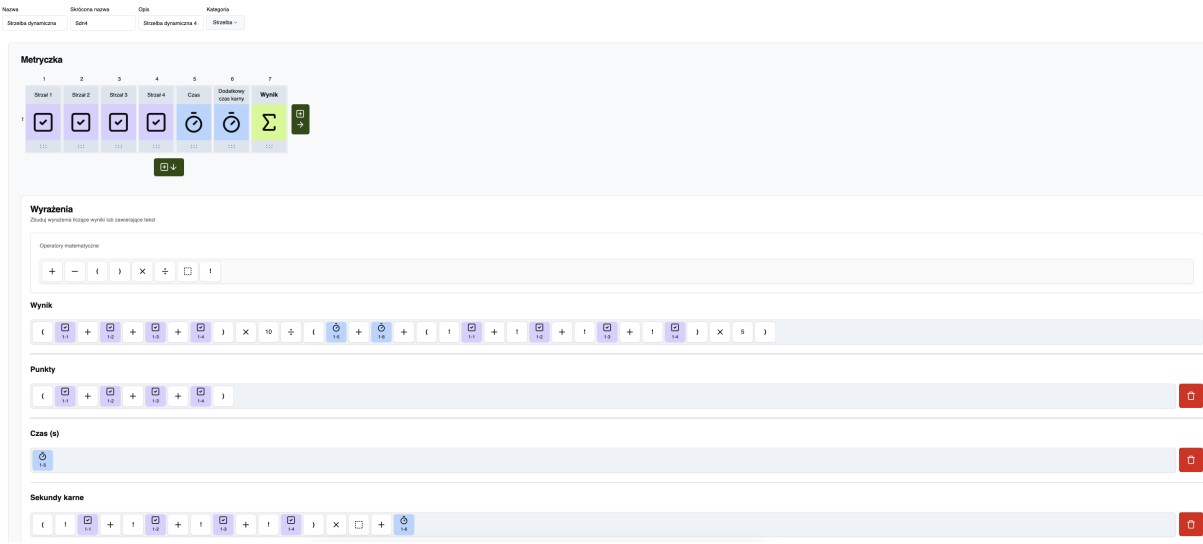

Rysunek 2: Zrzut ekranu z kreatora metryczki w systemie SCSM [Źródło: opracowanie własne, system SCMS].

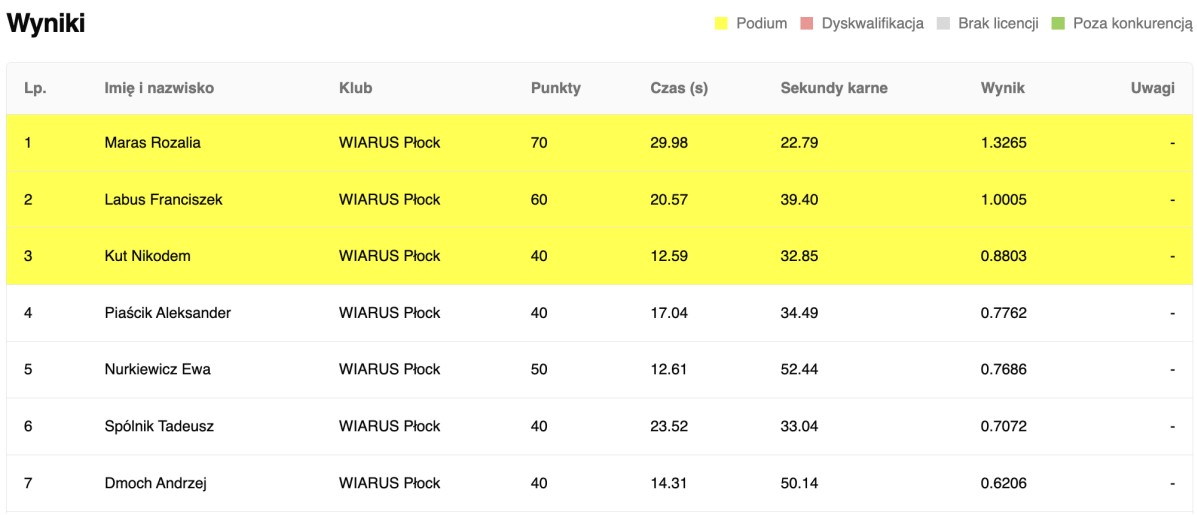

Rysunek 3: Zrzut ekranu z tabeli z wynikami na zawodach z utworzonej konkurencji z Rysunku 2 [Źródło: opracowanie własne, system SCMS].

- Organizacja zawodów:
  - Zarządzanie podstawowymi informacjami o zawodach
  - Konfiguracja konkurencji w ramach zawodów
  - System przydziału i zarządzania uprawnieniami sędziów

- Innowacyjny system obsługi uczestników:
  - Automatyczny system rezerwacji miejsc na zawodach
  - Generator metryczek z kodami QR dla każdej konkurencji, dla każdego zawodnika
  - Prosty system wprowadzania wyników zawodnika, który opiera się na skanowaniu kodu QR z metryczki i wprowadzeniu danych ustalonych w kreatorze metryczki (Rysunek 2)

- Zaawansowany system generowania oficjalnego raportu po zawodach strzeleckich, który zawiera takie dane jak podstawowe informacje o zawodach, obsadę sędziowską, tabelę z rankingiem dla każdej konkurencji. System automatycznie generuje raport w formacie DOCX na podstawie utworzonego wcześniej przez klub szablonu. Szablon to również plik w formacie DOCX, który może być stylizowany w dowolny sposób. System umożliwia dodanie do szablonu różnych zmiennych, które podczas generowania raport są podmieniane pod dane z zawodów. Dostępne zmienne to nazwa zawodów, data zawodów, lista konkurencji, lista z sędziami (w tym funkcja sędziego, konkurencje,

w których sędziuje, nazwisko, imię, numer licencji sędziowskiej i klasa sędziowska) oraz tabela z wynikami. Dzięki generowaniu raportu w formacie DOCX administrator, w razie potrzeby, w prosty sposób może poprawić wygenerowany raport i szybko wyeksportować go do formatu PDF, aby następnie udostępnić go jako oficjalny raport po zawodach.

- Spersonalizowane statystyki z wynikami dla zawodników

## 3.5   Porównanie systemów obsługi zawodów

Jednym z kluczowych osiągnięć projektu jest znacząca optymalizacja procesu organizacji i obsługi zawodów strzeleckich. Poniżej przedstawiamy szczegółowe porównanie tradycyjnego podejścia z naszym rozwiązaniem wykorzystującym system SCMS:

**Etap rejestracji:**

- **Metoda tradycyjna:**
    - Komunikacja telefoniczna lub SMS-owa z każdym zawodnikiem
    - Ręczne prowadzenie harmonogramu rezerwacji
    - Manualne wypełnianie metryczek dla każdej konkurencji

- **System SCMS:**
    - Automatyczny system zapisów online
    - Inteligentny harmonogram rezerwacji
    - Masowe generowanie metryczek z unikalnymi kodami QR

**Obsługa w dniu zawodów:**

- **Metoda tradycyjna:**
    - Dystrybucja indywidualnych metryczek papierowych
    - Fizyczny przepływ dokumentów między stanowiskami
    - Ręczne obliczanie wyników przez sędziów
    - Manualne tworzenie rankingów w arkuszu kalkulacyjnym

- **System SCMS:**
    - Zoptymalizowany format metryczek
    - Skanowanie kodów QR lub wprowadzanie 5-znakowego kodu
    - Automatyczne przeliczanie wyników
    - Generowanie rankingów w czasie rzeczywistym

**Dokumentacja pozawodowa:**

- **Metoda tradycyjna:**
    - Czasochłonne tworzenie raportu końcowego
    - Ręczne wprowadzanie wszystkich danych i wyników

- **System SCMS:**
    - Automatyczne generowanie raportów w formacie DOCX
    - Zaawansowany system personalizacji szablonów raportów
    - Szybka konwersja do formatu PDF

Bezpośrednie porównanie procesów przeprowadzania zawodów strzeleckich znajduje się w Tabeli 1.

| System tradycyjny | Shooting Club Management System |
| --- | --- |
| Manualna rejestracja telefoniczna lub SMS-owa | Automatyczny system zapisów online z harmonogramowaniem |
| Ręczne wypełnianie pojedynczych metryczek dla każdej konkurencji | Zautomatyzowane generowanie metryczek (Rysunek 4) z kodami QR |
| Ręczne obliczenia i wprowadzanie wyników na fizycznej metryczce | Automatyczne przeliczanie wyników po wprowadzeniu danych do wirtualnej metryczki |
| Manualnie tworzone rankingi w arkuszu kalkulacyjnym | Automatyczne generowanie rankingów w czasie rzeczywistym |
| Czasochłonne tworzenie raportów od podstaw | Automatyczne generowanie raportów z wykorzystaniem personalizowanych szablonów |

Tabela 1: Porównanie kluczowych funkcjonalności systemów.

Wizualne porównanie tradycyjnych metryczek z metryczkami wygenerowanymi przez SCMS:

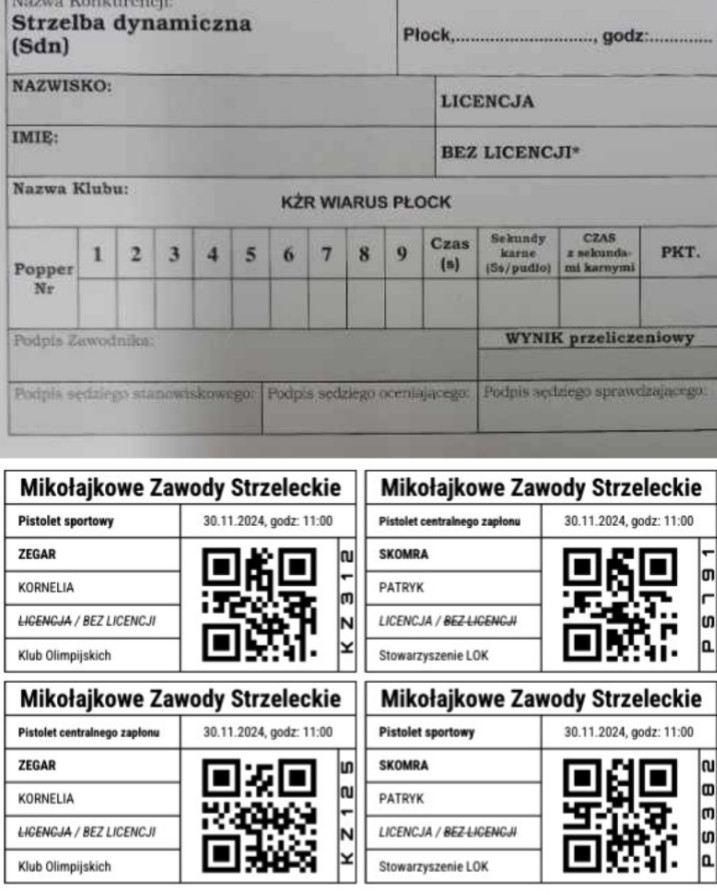

Rysunek 4: Porównanie aktualnych metryczek z metryczkami wygenerowanymi przez system SCMS [Źródło: opracowanie własne].

### 3.5.1 Innowacyjne rozwiązania

System wprowadza szereg unikalnych funkcjonalności:

- **Uniwersalny system tworzenia szablonów konkurencji**
  - Łatwy sposób tworzenia szablonu konkurencji oraz wyglądu wirtualnych metryczek
  - Pełna automatyzacja wyliczania wyniku zawodnika
  - Możliwość definiowania dodatkowych pól informatywnych dla konkurencji

- **Inteligentne metryczki z kodami QR**
  - Optymalizacja miejsca (4 metryczki na 1 stronie A5) - co przedstawiono na Rysunku 4.
  - Szybka identyfikacja poprzez skanowanie kodów QR
  - Alternatywny system kodów 5-znakowych
- **Zaawansowany system raportów**
  - Pełna personalizacja wyglądu dokumentów
  - System zmiennych do automatycznego wypełniania
  - Elastyczne dostosowanie do potrzeb klubu

# 4 WNIOSKI

Projekt dostarczył kompleksowe narzędzie wspierające zarządzanie klubami strzeleckimi w Polsce. Zapewnia cyfryzację oraz usprawnienie procesów administracyjnych i organizacyjnych, co pozwala znacząco zoptymalizować zarządzanie zawodami i treningami. W efekcie system może się przyczynić do poprawy jakości obsługi członków klubów oraz zwiększenia efektywności operacyjnej klubów.

W trakcie realizacji projektu udało się stworzyć produkt, który jest gotowy do wdrożenia w rzeczywistych warunkach. Cały system został zaprojektowany z myślą o użytkownikach końcowych, z uwzględnieniem ich potrzeb i specyfiki pracy klubów strzeleckich. Po przeprowadzeniu testów oraz weryfikacji działania systemu, aplikacja spełnia wszystkie wymagania funkcjonalne i jest gotowa do wdrożenia w środowisku produkcyjnym. Została również dostosowana do realnych warunków operacyjnych, co zapewnia jej stabilność i wydajność.

Dodatkowo skonfigurowano środowisko produkcyjne oraz wykupiono domenę, co umożliwia szybkie uruchomienie systemu. Wdrożono również zaawansowany system monitorowania aplikacji, pozwalający na bieżące śledzenie jej działania, analizę wydajności oraz szybkie reagowanie na ewentualne problemy. Rozwiązanie to wspiera utrzymanie wysokiej dostępności i niezawodności aplikacji.

Najważniejszym osiągnięciem projektu jest opracowanie kreatora konkurencji na zawody, który stanowi kluczowy element modułu organizacji zawodów. Narzędzie to umożliwia klubom w prosty i intuicyjny sposób tworzenie własnych konkurencji, które mogą być następnie efektywnie wykorzystywane podczas zawodów, co znacząco zwiększa elastyczność i funkcjonalność aplikacji.

# 5 KIERUNKI ROZWOJU

Kierunki rozwoju systemu Shooting Club Management System obejmują szereg zaplanowanych funkcjonalności, które mają na celu lepsze dopasowanie do potrzeb klubów strzeleckich oraz ich członków. W planach rozwojowych znajduje się implementacja systemu płatności online, który umożliwi automatyczne rozliczanie opłat za treningi i zawody, integrację z popularnymi systemami płatności, takimi jak BLIK czy przelewy, automatyczne generowanie faktur i potwierdzeń oraz dostęp do panelu raportowania finansowego.

Kolejnym kierunkiem rozwoju jest stworzenie forum klubowego, które posłuży jako platforma wymiany doświadczeń między członkami, system ogłoszeń klubowych oraz miejsce do tworzenia grup tematycznych i dyskusyjnych. Obecnie fora internetowe cieszą się dużą popularnością w społeczności strzeleckiej, a użytkownicy chętnie korzystają z nich w celach takich jak wymiana informacji, organizacja wydarzeń czy kupno i sprzedaż broni oraz akcesoriów strzeleckich. Dlatego wdrożenie takiej funkcji w systemie może znacząco zwiększyć zaangażowanie społeczności klubowej.

Dodatkowo planuje się stworzenie aplikacji mobilnej, która umożliwi szybki dostęp do funkcji takich jak rezerwacje, wyniki czy powiadomienia, a także wdrożenie systemu zarządzania sprzętem, który pozwoli na ewidencję i rezerwację wyposażenia klubowego. Rozwój systemu uwzględnia również integracje zewnętrzne, obejmujące połączenie z systemami związków strzeleckich oraz innych klubów.

# 6 PODZIĘKOWANIA

Serdeczne podziękowania dla dr inż. Marcina Pietranika za wsparcie techniczne i merytoryczne, które miało kluczowe znaczenie dla realizacji projektu, oraz dla Klubu Żołnierzy Rezerwy Ligi Obrony Kraju w Płocku "WIARUS" za dostarczenie materiałów, które w istotny sposób przyczyniły się do osiągnięcia założonych celów.