# OpenReview forum: "Shooting Club Management System"
_pwr.edu.pl/Wrocław_University_of_Science_and_Technology/2024/ZPI_Day — Wrocław University of Science and Technology 2024 ZPI Day Submission_

### Official Review · Reviewer_3iLY · 2024-12-03
**Recenzja systemu SCMS**

**Confidence:** 5
**Significance Of Results:** 5
**Overall Quality:** 5

**Compliance With Template:**

5: Very High Quality – The article contains all the required sections, which are written in a very detailed, clear, and error-free manner. The structure is professional and meets expectations, and the content adheres to the highest substantive and formal standards.

**Description Of Results:**

5: Very High Quality – The results are described in detail, clearly and comprehensively, supported by thorough evaluation, analysis, and convincing usage examples. The description meets the highest substantive standards.

**Feedback On Consistency:**

Opis projektu cechuje się spójnością i logicznym układem treści. Zidentyfikowany problem – brak dedykowanych rozwiązań informatycznych w polskich klubach strzeleckich – został jasno przedstawiony. Cele projektu, obejmujące automatyzację procesów, poprawę organizacji zawodów i treningów oraz zwiększenie efektywności operacyjnej, są logicznie powiązane z przedstawionym problemem. Wyniki projektu, takie jak rozwój modułów funkcjonalnych systemu (zarządzanie klubem, licencjami, zawodami i treningami), korespondują z założonymi celami. Wnioski potwierdzają osiągnięcie założonych celów oraz wskazują na gotowość systemu do wdrożenia w realnych warunkach.

**Potential For Development:**

Artykuł wskazuje na liczne możliwości rozwoju systemu. Planowane funkcjonalności obejmują m.in. rozszerzenie systemu o moduł płatności online, co usprawni rozliczenia za treningi i zawody oraz wprowadzi dodatkowe opcje, takie jak automatyczne generowanie faktur i raportów finansowych. Wdrożenie forum klubowego jako platformy komunikacji dla członków klubów, wspierające organizację wydarzeń i wymianę doświadczeń, jest kolejnym krokiem zwiększającym zaangażowanie społeczności. Stworzenie aplikacji mobilnej, która umożliwi łatwy dostęp do funkcji takich jak rezerwacje czy powiadomienia, a także rozwój systemu zarządzania sprzętem, pozwolą na jeszcze dopasowanie systemu do codziennych potrzeb klubów strzeleckich.

**Project Nature Evaluation:**

Projekt reprezentuje cechy pracy inżynierskiej:
- System oferuje znaczną wartość praktyczną, upraszczając i automatyzując kluczowe procesy zarządzania klubem strzeleckim oraz organizacji zawodów. Propozycja usprawnień, takich jak generowanie raportów i wirtualnych metryczek, znacznie zwiększa efektywność i transparentność.
- Zastosowanie nowoczesnych technologii, takich jak Django, React oraz PostgreSQL, świadczy o zaawansowanym podejściu inżynierskim. Projekt wykorzystuje także technologie wspierające, takie jak Docker, Redis i Sentry, co dodatkowo podkreśla profesjonalizm techniczny.

**Technical Language Precision:**

5: Very High Quality – The language is entirely appropriate for a technical report. All terms are used correctly and precisely, and the style is professional, clear, and coherent, without any errors or ambiguities.

---

### Official Review · Reviewer_zd9u · 2024-12-06
**Shooting Club Management System - recenzja**

**Confidence:** 5
**Significance Of Results:** 5
**Overall Quality:** 5

**Compliance With Template:**

5: Very High Quality – The article contains all the required sections, which are written in a very detailed, clear, and error-free manner. The structure is professional and meets expectations, and the content adheres to the highest substantive and formal standards.

**Description Of Results:**

5: Very High Quality – The results are described in detail, clearly and comprehensively, supported by thorough evaluation, analysis, and convincing usage examples. The description meets the highest substantive standards.

**Feedback On Consistency:**

Abstrakt jest napisany w sposób spójny i pokrywa się z szablonem. Zasadniczo brakiem można doszukać się dwóch braków - skrótowiec "SCMS" nie został nigdzie wprowadzony tj. nie jest bezpośrednio określone jego znaczenie oraz brakuje sekcji bibliograficznej wyszczególnionej w szablonie abstraktu. Ta sekcja, wydaje się być jednak zbyteczna w kontekście pracy i dlatego nie ma wpływu na ocenę podczas recenzji. Można ją jednak dodać pro forma, aby zapewnić stuprocentową zgodność z szablonem.

**Potential For Development:**

Abstrakt zawiera rozwinięty opis potencjalnych ścieżek rozwoju, które wydają się stanowić naturalny krok ku rozwojowi aplikacji w cały, kompleksowy system "strzelecki" rozszerzony nie tylko na funkcjonowanie poszczególnych placówek pod kątem administracyjnym, ale też o aspekty społecznościowe.

**Project Nature Evaluation:**

Projekt ewidentnie spełnia wymagania stawiane przed pracą inżynierską poprzez rozwiązywanie typowego problemu za pomocą dostępnych, gotowych i adekwatnych rozwiązań. Bardzo zgrabnie opisano tło projektu, uzasadniono i podsumowano użycie wybranych technologii oraz podsumowano wyniki. Jedyną wątpliwość można by odnieść do metodologii projektowania baz danych. Mianowicie, przechowywanie danych w formacie JSON przypomina tworzenie atrybutu złożonego, co można by teoretycznie uznać za złamanie warunku 1-szej postaci normalnej, czyli atomowości atrybutów. Jak jednak wspomniano w samym abstrakcie, istotą tej decyzji było przechowywanie danych nieregularnych. Wydaje się, że można by zaprojektować bazę tak, aby w pełni wykorzystywać model relacyjny do zrealizowania tej funkcjonalności lub oprzeć rozwiązanie o podejście nierelcayjne, jednak jak to często bywa z zasadami - w praktyce niektóre z nich można świadomie łamać. Dlatego też, powyższa uwaga jest raczej sugestią, a nie zarzutem i nie ma wpływu na ocenę.

**Technical Language Precision:**

5: Very High Quality – The language is entirely appropriate for a technical report. All terms are used correctly and precisely, and the style is professional, clear, and coherent, without any errors or ambiguities.

---

### Official Review · Reviewer_L1qh · 2024-12-06
**Recenzja Shooting Club Management System**

**Confidence:** 4
**Significance Of Results:** 4
**Overall Quality:** 4

**Compliance With Template:**

3: Average Quality – The article includes most of the required sections, but some may be incomplete, written in a general or unclear manner. The content is correct but requires further refinement.

**Description Of Results:**

4: High Quality – The results are described in detail and supported by usage examples or evaluations. The description is reliable but may lack full depth of analysis.

**Feedback On Consistency:**

Analiza problemu, cel i zakres funkcjonalny sposób został przedstawiony w zwięzły i jasny sposób. Prezentacja wyników jest przedstawiona w kontekście zdefiniowanych problemów. Podsumowanie i proponowana dalsza praca na projektem wydaje się być logicznym następstwem uzyskanych rezultatów.

**Potential For Development:**

Projekt ma potencjał rozwoju a może i komercjalizacji.

**Project Nature Evaluation:**

Wybór technologii użytej do wytworzenia systemu wydaje się być właściwy, niestety autorzy słabo go uzasadniają. Niedostatecznie opisano przebieg procesu wytwórczego, poza wskazaniem problemów w jego trakcie. Nie opisano też w jasny sposób decyzji architektonicznych i projektowych.

**Technical Language Precision:**

3: Average Quality – The language is mostly appropriate but may contain minor terminological or stylistic errors. Some statements might lack precision or require improvement for better readability.

---

### Decision · Program_Chairs · 2024-12-10

Accept (Oral)